# Determinants of diabetic retinopathy in Ethiopia: A systematic review and meta-analysis

Abere Woretaw Azagew[1]*, Yeneabat Birhanu Yohanes[2], Zerko Wako Beko[1], Yohannes Mulu Ferede[1], Chilot Kassa Mekonnen[1]

1 Department of Medical Nursing, School of Nursing, College of Medicine and Health Sciences, University of Gondar, Gondar, Ethiopia, 2 Department of Surgical Nursing, School of Nursing, College of Medicine and Health Sciences, University of Gondar, Gondar, Ethiopia

* wabere@ymail.com

**Data Availability Statement:** All relevant data are within the paper and its Supporting Information files.

## Abstract

### Introduction

Diabetic retinopathy (DR) is the primary retinal vascular complication of diabetes mellitus and a leading cause of visual impairment and blindness. It affects the global diabetic population. In Ethiopia, about one-fifth of diabetic patients were affected by DR, but there were inconsistent finding across studies about the determinants factors of DR. Therefore, we aimed to identify the risk factors for DR among diabetic patients.

### Methods

We have accessed previous studies through an electronic web-based search strategy using PubMed, Google (Scholar), the Web of Science, and the Cochrane Library with a combination of search terms. The quality of each included article was assessed using the Newcastle Ottawa Assessment Scale. All statistical analyses were carried out using Stata version 14 software. The odds ratios of risk factors were pooled using a fixed-effect meta-analysis model. Heterogeneity was assessed using the Cochrane Q statistics and I-Square ($I^2$). Furthermore, publication bias was detected based on the graphic asymmetry test of the funnel plot and/or Egger's test ($p < 0.05$).

### Results

The search strategy retrieved 1285 articles. After the removal of duplicate articles, 249 articles remained. Following further screening, about 18 articles were assessed for eligibility, of which three articles were excluded because of reporting without the outcome of interest, poor quality, and not full text. Finally, fifteen studies were reviewed for the final analysis. Co-morbid hypertension (HTN) (AOR 2.04, 95%CI: 1.07, 3.89), poor glycemic control (AOR = 4.36, 95%CI: 1.47, 12.90), and duration of diabetes illness (AOR = 3.83, 95%CI: 1.17, 12.55) were found to be confirmed associated factors of diabetic retinopathy.

**Funding:** The author(s) received no specific funding for this work.

**Competing interests:** The authors have declared that no competing interests exist.

**Abbreviations:** AOR, Adjusted Odds Ratio; CI, Confidence Interval; DR, Diabetic retinopathy; HbA1c, Glycated hemoglobin; HTN, Hypertension; OR, Odds Ratio; PRISMA, Preferred Reporting Item of Systematic Review and Meta-Analysis; SNNP, Southern Nations Nationalities and People.

## Conclusion

In this study, co-morbid HTN, poor glycemic control, and longer duration of diabetes illness were found to be the determinant factors of DR. Aggressive treatment of co-morbid HTN and blood glucose, and regular eye screening should be implemented to reduce the occurrence of DR.

## Trial registration

The review protocol was registered in the international prospective register of systematic reviews (PROSPERO) with registration number PROSPERO: CRD42023416724.

## Introduction

Diabetic retinopathy is a diabetic complication that affects the eyes. It is caused by damage to the blood vessels of the light-sensitive tissue at the back of the eyes [1]. Diabetic retinopathy is recognized as microvascular complication [2]. Clinically, diabetic retinopathy is classified as non-proliferative and proliferative diabetic retinopathy. Non-proliferative diabetic retinopathy (NPDR) is the earliest and the asymptomatic stage whereas proliferative diabetic retinopathy (PDR) is the advanced stage of diabetic retinopathy characterized by neovascularization. In the PDM stage, the patient experiences severe vision impairment when the new abnormal blood vessel bleeds to the vitreous (vitreous hemorrhage) or retinal detachment [3]. The most common cause of vision loss in patients with diabetic retinopathy is diabetic macular edema. It is characterized by swelling or thickening of the macula which causes distortion of visual images and a decrease in visual acuity [4, 5].

Diabetic retinopathy is the leading cause of visual impairment or blindness among the working-age population in the world. It is considered the third leading cause of blindness worldwide [6]. It affects both type 1& 2 diabetic patients, but its incidence is higher in type 1 than in type 2 diabetes mellitus populations [7].

The global prevalence of DR was (22.27%) [8], Africa (33.8%) [6], and Ethiopia (19.48%) [9]. The pathogenesis of diabetic retinopathy is not well known, but it is associated with high blood glucose. Overtime, having too much sugar in the blood can damage the retina vasculature. In the earlier stage, it forms microaneurysm, capillary leakage, retinal edema, capillary occlusion, ischemia, and cotton wool spot formation. If it is not treated, it results in loss of vision [10, 11].

Even though prevention strategies such as keeping blood glucose to the optimum level, adopting a healthy lifestyle, losing weight, exercising regularly, and health education to adhere to medications [12, 13] implemented in diabetic follow-up care to reduce the incidence of diabetic retinopathy, but the problem is still rising [14]. Therefore, risk stratification and selective early intervention for high-risk patients need to be given attention. In Ethiopia, there were different research articles reporting on the determinant factors of diabetic retinopathy, but their findings were inconsistent across the studies. Therefore, this study aimed to identify the determinant factors of DR among diabetic patients.

## Methods

### Reporting

The review protocol has been registered in the international prospective register of systematic reviews (PROSPERO) with registration number (PROSPERO: CRD42023416724), and the

result of the review was presented based on standard Preferred Reporting Items for Systematic review and Meta-analysis (PRISMA) [15] checklist (**S1 File**).

## Study selection and search strategy

The procedure for this systematic review and meta-analysis was designed following the Preferred Reporting Items for Systematic Review and Meta-Analysis (PRISMA) flow chart [15]. We searched on PubMed, Google (for grey literature), Google Scholar, Web of Science, and Cochrane Library databases for studies reporting diabetic retinopathy. Endnote (Version 7) reference management software was used to download, organize, review, de-duplicate, and cite the articles. Our comprehensive search strategies were carried out using controlled vocabularies (MeSH terms). Using the MeSH database, the synonyms of diabetic retinopathy were identified. Then, the search string was established using the databases. Articles were searched by title (Ti), abstract (Ab), full text, or all these categories. Modification of the search strategy was made by limiters such as study design, and country. Boolean logic operators such as "AND" and "OR" were used to combine searching terms. The search strings were stated as: "diabetic retinopathy" OR "diabetic retinopathy*" OR "diabetic eye complication" OR "diabetic macular edema" OR diabetic macular? edema OR "diabetic angiopathy" OR "diabetic angiopath?" AND "adult diabetic patients" OR "diabetes mellitus patients" AND Ethiopia. Two reviewers independently searched and screened articles by title, abstract, and full text. The disagreements between the reviewers were resolved by discussion.

## Inclusion and exclusion criteria

The eligibility of the included studies was summarized in the table below (**Table 1**).

## Data extraction

The data were extracted by data abstraction format using the Microsoft excel spreadsheet. The format was developed by two reviewers and piloted for its clarity, aim, consistency, and depth of the contents. Simple and consistent codes of response were used. Then the reviewers independently reviewed and extract data from each eligible study. The information such as authors, publication year, region of the study, design, methodological quality, population, study setting, sample size, method of data collection, statistical analysis, and funding source were extracted from the studies (**S2 File**).

**Table 1. Inclusion and exclusion criteria for included research articles.**

| Criteria | Inclusion criteria | Exclusion criteria |
|---|---|---|
| Participants | People with diabetic retinopathy age ≥18 years | Population with no outcome interest |
| Study setting | Hospital or health facility | Community-based study |
| Design | Observational study designs (cross-sectional, cohort, and case-control) | |
| Publication status | both published and unpublished studies | Qualitative studies conference papers articles with no full text |
| Language of publication | English | Languages other than the English language |
| Country | Ethiopia different regions of the country | |
| Publication year | No restriction | |

### Quality assessment

Articles were assessed for quality score using the New Castle Ottawa Scale adapted from cross-sectional, cohort, and case control' quality assessment tools; a score of $\geq 7$ out of 10 was considered a high-quality score [16]. Two reviewers (CKM and HMA) assessed the quality of each paper. The reviewers compared the quality of the appraisal scores and resolved inconsistencies before calculating the final appraisal score. All the included studies had the high-quality scores. The PRISMA checklist 2020 [17] was used to report the results of this study.

### Data analysis

The data were entered to a Microsoft excel spreadsheet and exported to Stata version 14 for analysis. Cochran's Q statistic and I-squared ($I^2$) were used to evaluate the presence of heterogeneity. The I-square test statistic results of 25%, 50%, and 75% were declared as low, moderate, and high heterogeneity [18], respectively. The pooled summary effect size was estimated using the fixed effect model [19]. The publication bias was detected based on the graphic asymmetry test of the funnel plot and/or Egger's test ($p < 0.05$) [20].

## Results

### Study selection and characteristics

The search strategy retrieved 1285 articles. After the removal of duplicate articles, 249 articles remained. Following the additional screening, eighteen articles were evaluated for eligibility, with three being excluded due to being incomplete, of poor quality, or not in full text (**Fig 1**). Finally, fifteen studies were reviewed for co-morbid HTN [21–35] (**Table 2**), six studies for poor glycemic control [21, 22, 24, 25, 28, 32] (**Table 3**), and five studies for the duration of diabetic illness [21, 22, 25, 27, 31] (**Table 4**). Of the total studies; five were conducted in the Amhara region [21, 23, 27, 30, 34], five in Addis Ababa [24, 26, 29, 32, 35], three in the Oromia region [25, 31, 33], and two in South Nations and Nationalities people (SNNP) of Ethiopia [22, 28].

All the studies were published between the years 2015 and 2022. Regarding the study design, eight studies were cross-sectional [21–23, 26, 27, 32, 34, 35], five studies were cohort [28–31, 33], and two studies were case-control [24, 25]. The overall quality score of the included studies' was $\geq 7$.

## Determinants factors of diabetic retinopathy

### Co-morbid hypertension

In this systematic review and meta-analysis, co-morbid HTN is found to be the determinant factor for diabetic retinopathy. Diabetic patients who have co-morbid HTN are 2.04 times more likely to have diabetic retinopathy compared to those diabetic patients with no co-morbid HTN (AOR 2.04, 95%CI: 1.07, 3.89) (**Fig 2**).

### Heterogeneity and publication bias of included studies

The overall heterogeneity test ($I^2$) on the effect of co-morbid HTN was 0.0% with a p-value < 0.946, using a random effect model to adjust observed variability. This indicates there is no variability across the studies.

Regarding the publication bias, the graphic asymmetry test of the funnel plot which shows a symmetrical distribution (**Fig 3**), and Egger's test p-value = 0.181, indicating that there is no publication bias.

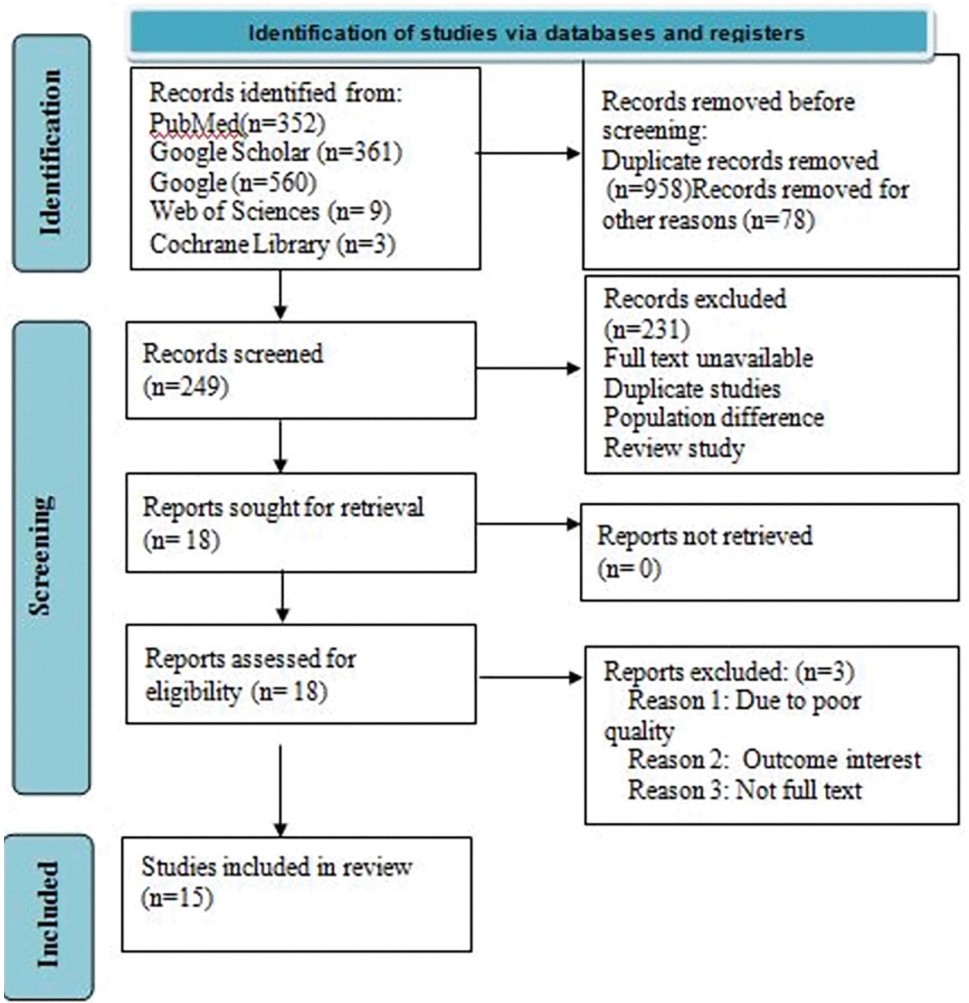

**Fig 1. PRISMA flow chart for flow of information through the phase of systematic review.**

## Poor glycemic control

In this study, poor glycemic control was found to be the determinant factor for diabetic retinopathy. Diabetic patients with poor glycemic control status were 4.36 times more likely to have diabetic retinopathy compared to those diabetic patients with good glycemic control status (AOR = 4.36, 95%CI: 1.47, 12.90) (**Fig 4**).

## Heterogeneity and publication bias of included studies

The heterogeneity test ($I^2$) on the effect of poor glycemic control was 0.0% with a p value< 0.847, using a random effect model to adjust observed variability. This indicates there is no variability across the studies. The publication bias was assessed using the graphic asymmetry test of the funnel plot, shows a symmetrical distribution (**Fig 5**), and Egger's test p value = 0.851means that there was no publication bias.

## Duration of diabetic illness

Longer duration of diabetic illness is found to be the risk factor for the development of diabetic retinopathy. Patients who had a longer duration of diabetic illness were nearly four times more

**Table 2. Study characteristics by co-morbid HTN among diabetic patients in Ethiopia.**

| Author/year | Study design | Region of the study | data collection technique | Funding Source | Factors | AOR | 95% CI | overall quality score |
|---|---|---|---|---|---|---|---|---|
| Tilahun M, et al./2020 [21] | Cross-sectional | Amhara | Interview & patient review | not funded | Co-HTN | 3.39 | 1.64–7.02 | 8.5 |
| Alemayehu HB, et al./2022 [22] | Cross-sectional | SNNP | Interview& patient chart review | not reported | Co-HTN | 1.43 | 0.72–2.86 | 7.4 |
| Mersha GA, et al./2021 [23] | Cross-sectional | Amhara | interview & chart review | not reported | Co-HTN | 1.67 | 0.66–4.20 | 7 |
| Seid K, et al./2021 [24] | case-control | Addis Ababa | interview & chart review | Jimma University, Institute of Health | Co-HTN | 12.3 | 6.95–21.8 | 7.5 |
| Garoma D, et al./2020 [25] | Case-control | Oromia | interview & chart review Ocular exam | Jimma University, Institute of Health | Co-HTN | 3.38 | 1.29–9.05 | 7.0 |
| Shiberu T, et al./2018 [26] | Cross-sectional | Addis Ababa | interview & chart review Ocular exam | Not reported | Co-HTN | 2.556 | 1.014–6.447 | 9.0 |
| Ejigu T, et al./2021 [27] | Cross-sectional | Amhara | interview & chart review Ocular exam | Not reported | Co-HTN | 2.65 | 1.02–6.87 | 8.0 |
| Chisha Y, et al./2017 [28] | Cohort | SNNP | Record review | Mekelle University | Co-HTN | 4.1 | 1.76–9.44 | 7.5 |
| Azeze TK, et al./2018 [29] | Cohort | Addis Ababa | Record review | self sponsored | Co-HTN | 1.51 | 0.48–4.74 | 8.5 |
| Takele MB, et al./2022 [30] | Cohort | Amhara | Record review | Amhara regional state | Co-HTN | 1.68 | 1.14–2.50 | 8.4 |
| Gelcho GN, et al./2022 [31] | Cohort | Oromia | Record review | Not Funded | Co-HTN | 2.32 | 1.12–4.39 | 7.6 |
| Aberra T, et al./2022 [32] | Cross-sectional | Addis Ababa | Interview Record review | Not reported | Co-HTN | 1.37 | 0.865–2.169 | 8.5 |
| Debele GR, et al./2021 [33] | Cohort | Oromia | Record review | University of Gondar | Co-HTN | 0.54 | 0.35–0.82 | 8.0 |
| Alemu S, et al./2015 [34] | Cross-sectional | Amhara | Interview & record review | not reported | Co-HTN | 5.2 | 2.5–10.20 | 7.5 |
| Abera F, et al./2021 [35] | Cross-sectional | Addis Ababa | interview & chart review ocular exam | Not reported | Co-HTN | 8.63 | 2.51–29.75 | 7.0 |

**Notes**; AOR; Adjusted odds ratio, CI; Confidence Interval, Co-HTN: Co-morbid Hypertension, SNNP: Southern Nations, Nationalities and Peoples'

likely to have diabetic retinopathy compared to those diabetic patients with a shorter duration of diabetic illness (AOR = 3.83, 95%CI: 1.17, 12.55, $I^2$ = 0.0% and p-value = 0.999) (**Fig 6**).

## Heterogeneity and publication bias of included studies

The overall heterogeneity test ($I^2$) on the effect of co-morbid hypertension was 0.0% with a p-value< 0.999, using a random effect model to adjust observed variability. This indicates there

**Table 3. Study characteristics by poor glycemic control status among diabetic patients in Ethiopia.**

| Author/year | Study design | Region of the study | Factors | AOR | 95% CI |
|---|---|---|---|---|---|
| Tilahun M, et al./2020 [21] | Cross-sectional | Amhara | PGC | 4.58 | 1.86–11.31 |
| Alemayehu HB, et al./2022 [22] | Cross-sectional | SNNP | PGC | 4.34 | 2.26–8.34 |
| Mersha GA, et al./2021 [23] | Cross-sectional | Amhara | PGC | 3.2 | 1.5–6.7 |
| Seid K, et al./2021 [24] | case-control | Addis Ababa | PGC | 10.7 | 6.17–18.58 |
| Garoma D et al./2020 [25] | Case-control | Oromia | PGC | 9.08 | 3.7–22.29 |
| Aberra T, et al./2022 [32] | Cross-sectional | Addis Ababa | PGC | 1.23 | 0.667–2.276 |

**Notes;** AOR; Adjusted odds ratio, CI; Confidence Interval, PGC: Poor glycemic control, SNNP: Southern Nations, Nationalities and Peoples'

**Table 4. Study characteristics by duration of diabetic illness among diabetic patients in Ethiopia.**

| Author/year | Study design | Region of the study | Factors | AOR | 95% CI |
|---|---|---|---|---|---|
| Tilahun M, et al./2020 [21] | Cross-sectional | Amhara | Duration Diabetic illness | 3.91 | 1.86–8.23 |
| Alemayehu HB, et al./2022 [22] | Cross-sectional | SNNP | Duration Diabetic illness | 4.78 | 2.11–10.83 |
| Garoma D, et al./2020 [25] | Case-control | Oromia | Duration Diabetic illness | 4.38 | 2.65–7.22 |
| Ejigu T, et al./2021 [27] | Cross-sectional | Amhara | Duration Diabetic illness | 2.91 | 1.01–8.35 |
| Gelcho GN, et al./2022 [31] | Cohort | Oromia | Duration Diabetic illness | 2.86 | 1.41–5.31 |

**Notes;** AOR; Adjusted odds ratio, CI; Confidence Interval, PGC- Poor glycemic control, SNNP: Southern Nations, Nationalities and Peoples'

was no variability across the studies. Concerning the publication bias, the graphic asymmetry test of the funnel plot shows a symmetrical distribution (**Fig 7**), and Egger's test p-value = 0.624, indicating that there is no publication bias.

## Discussion

Diabetic patients experience different complications along the continuum of their lives, of which diabetic retinopathy is the most common. We attempted to investigate the determinant factors of diabetic retinopathy. In this systematic review and meta-analysis, co-morbid HTN, poor glycemic control status, and longer duration of diabetic illness were found to be the determinate factors of diabetic retinopathy.

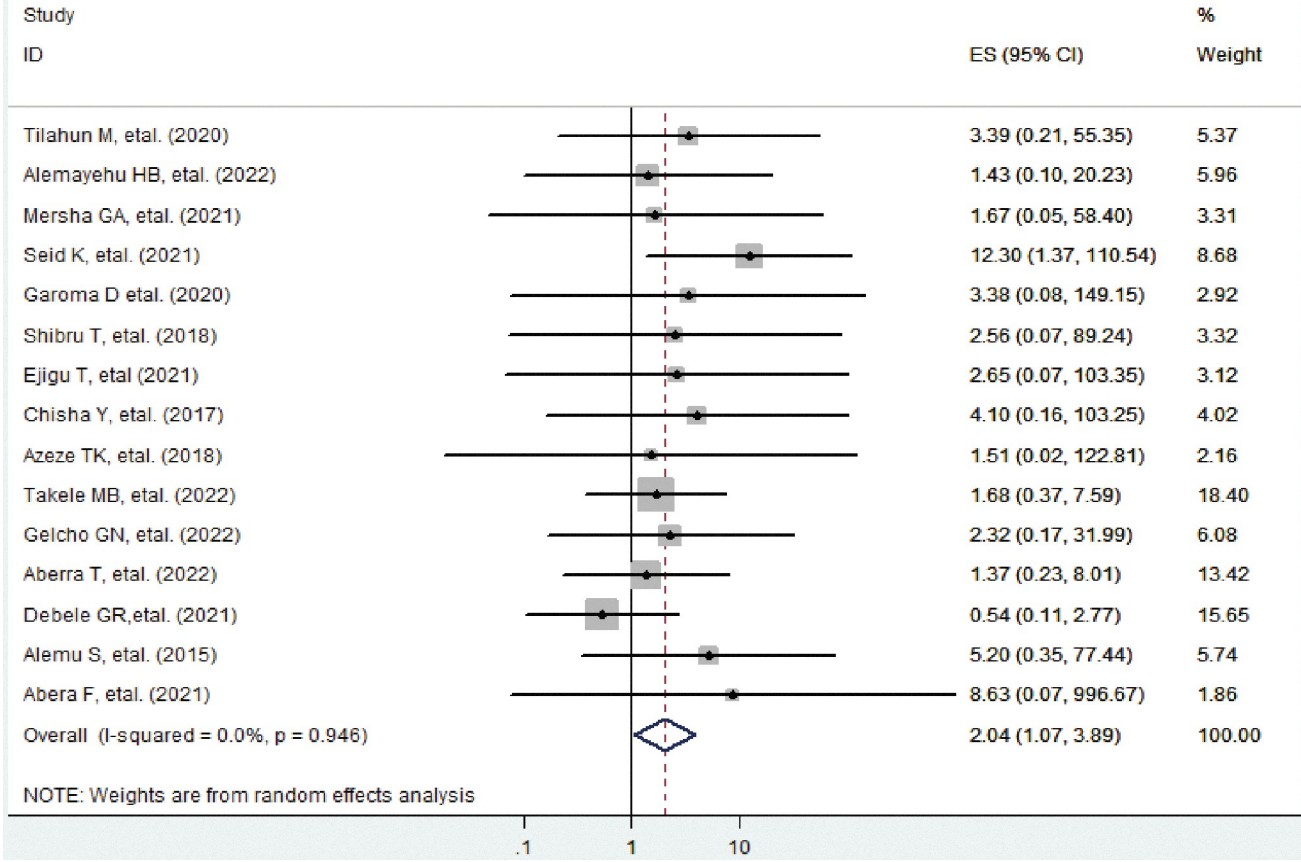

**Fig 2. Forest plot shows the pooled effect of co-morbid hypertension on diabetic retinopathy.**

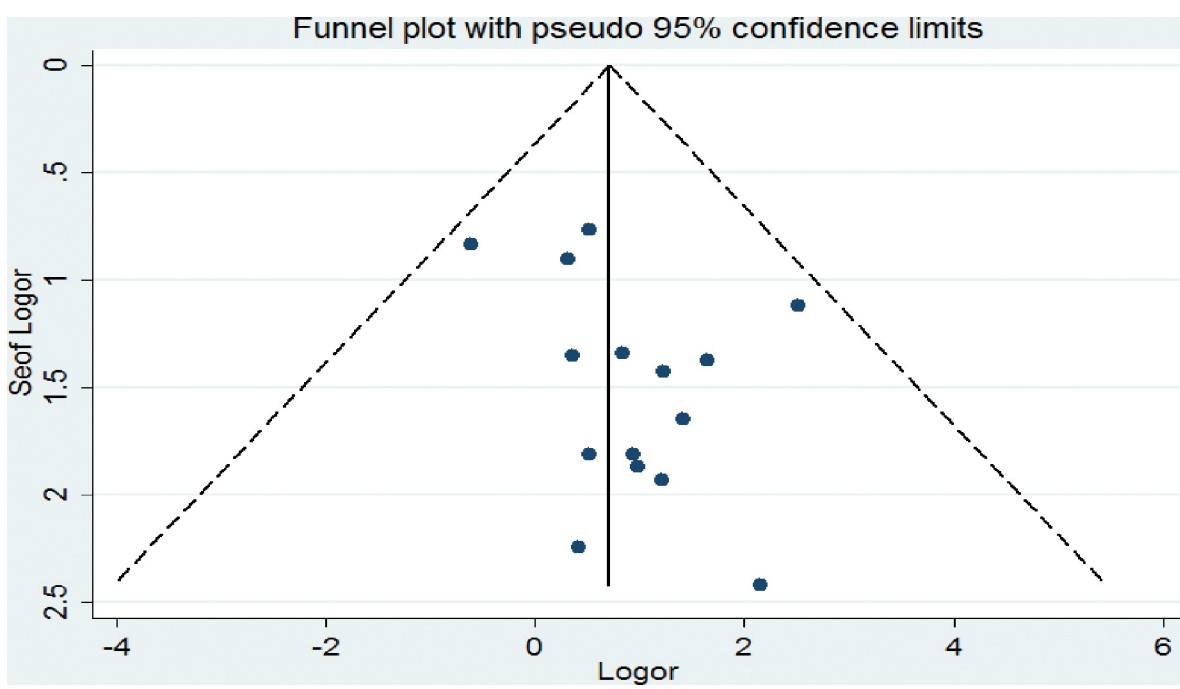

**Fig 3. Funnel plot to assess the heterogeneity of the included studies.**

The pooled effect of this study depicts that diabetic patients who have co-morbid HTN were 2.04 times more likely to have diabetic retinopathy compared to those diabetic patients with no co-morbid HTN. The finding of this study is supported by large-scale studies in China [36], and (Chinese, Malay, and India) [37]. This is the fact that HTN has a direct impact on retinal blood vessels. It damages the retinal vascular structures [38]. The elevated blood pressure

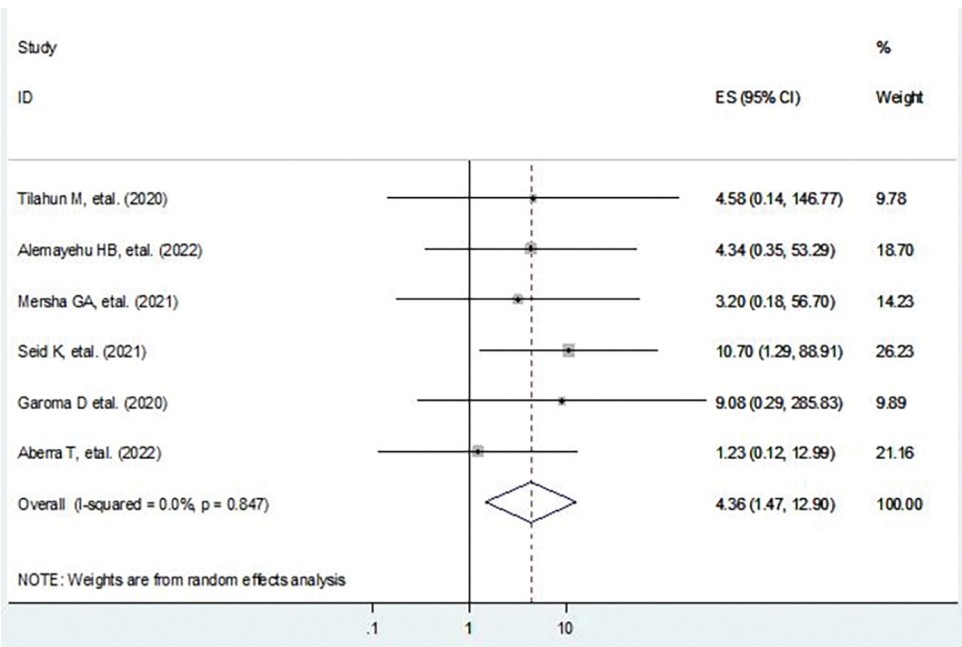

**Fig 4. Forest plot shows the pooled effect of poor glycemic control on diabetic retinopathy.**

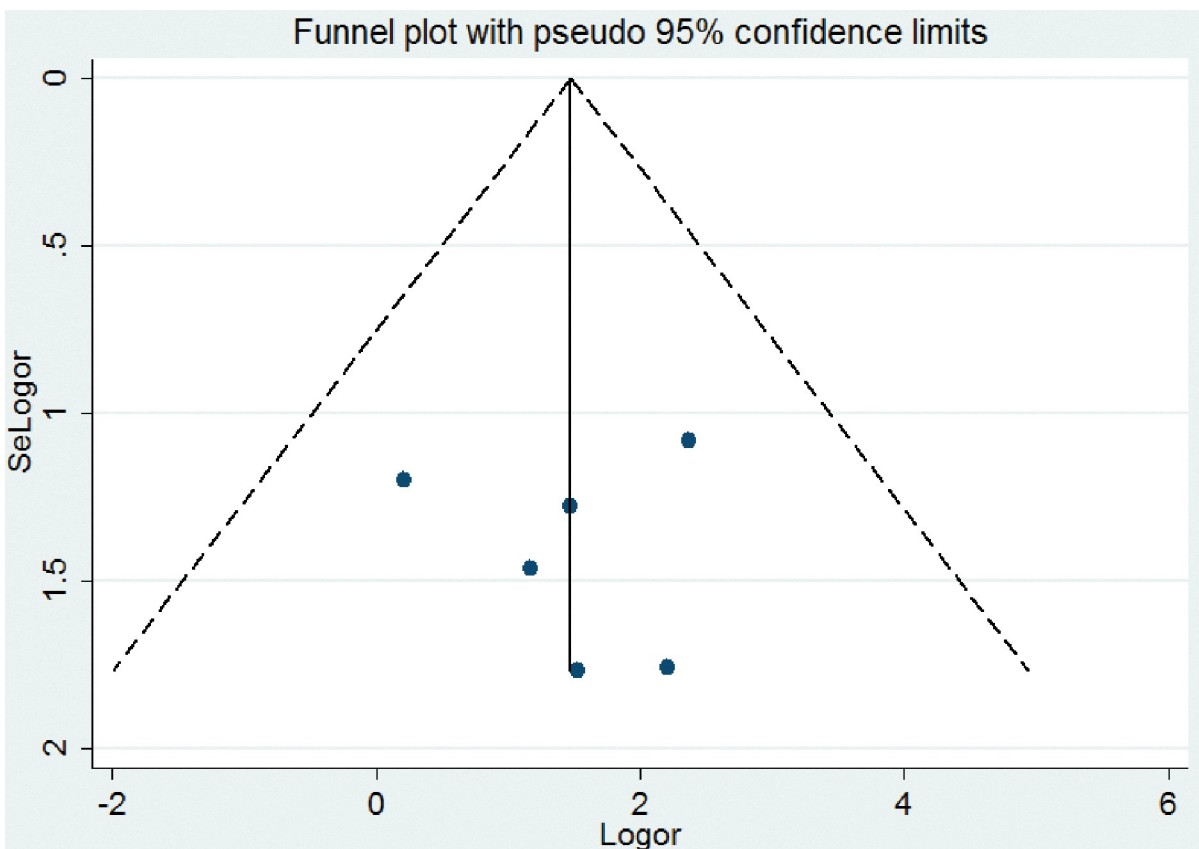

**Fig 5. Funnel plot to assess the heterogeneity of the included studies.**

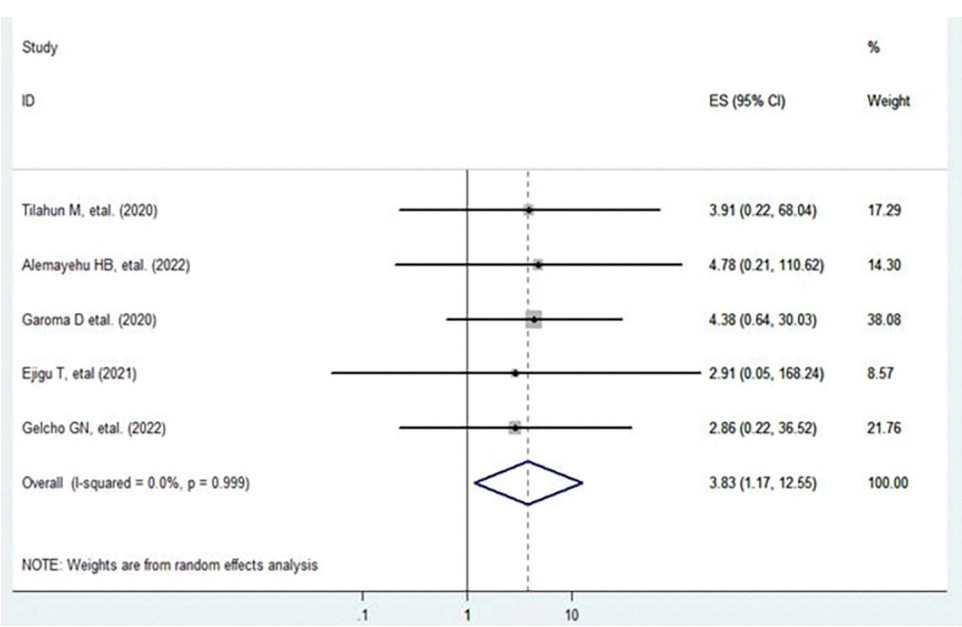

**Fig 6. Forest plot shows the pooled effect of duration of diabetic illness on diabetic retinopathy.**

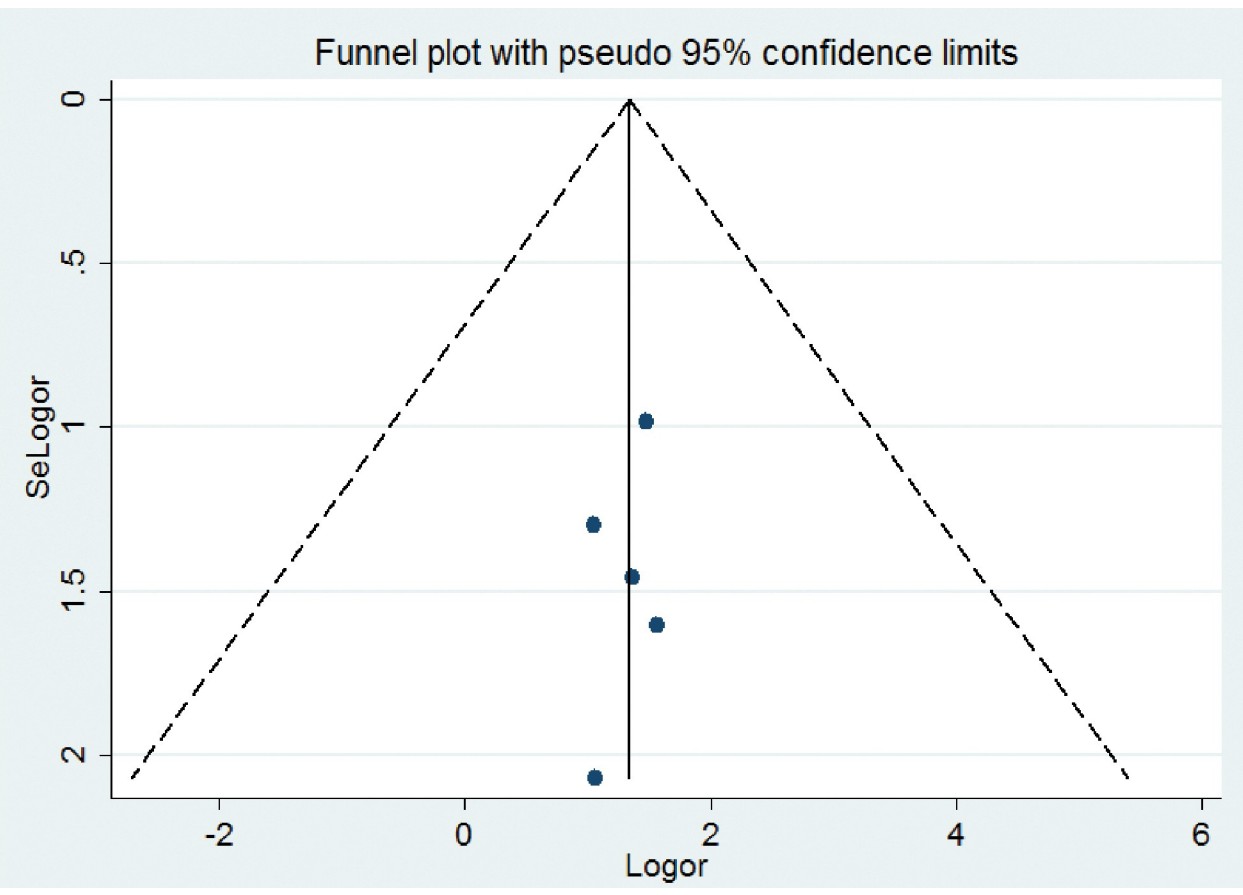

**Fig 7. Funnel plot to assess the heterogeneity of the included studies.**

is transferred directly to the vessels, which initially constrict, but a further increase in BP overcomes this compensatory tone, and damage to the muscle layer and endothelium ensues [39]. This results in retinal edema, cottonwool spots, hemorrhage, and disc edema [40]. Evidence showed that tight blood pressure control in the diabetic population reduces the incidence of sight-threatening retinopathy with a favorable impact on the lives of diabetic patients. A decrease in every 10 mmHg of blood pressure leads to are duction in 35% of retinopathy, a 35% need for retinal laser, and 50% blindness [41].

Similarly, poor glycemic control status was the determinant factor for diabetic retinopathy. Diabetic patients who have poor glycemic control status were nearly 4.4 times more likely to develop diabetic retinopathy compared to those diabetic patients with good glycemic control status. The finding of this study is supported by the study conducted in China [36, 42]. The possible reason is that too much blood glucose in the blood may block the tiny blood vessels that nourish the retina. As a result, the eye attempts to grow new blood vessels, but these new blood vessels do not develop properly and can leak easily, which leads to vascular edema [43]. Living with high blood glucose is the trigger for retinal vascular structure abnormality. The endothelial cells' malfunction owing to chronic exposure to high levels of glucose leads to endothelial cell malfunction. The resulting lesions include thickened capillary basement membrane, defects in the blood-retinal barrier, and pericyte loss [44].

Furthermore, living longer time with diabetes was a risk factor for diabetic retinopathy. Patients who have a longer duration of diabetic illness were nearly four times more likely to

have diabetic retinopathy compared to those diabetic patients with a short duration of diabetic illness. This is supported by the study conducted in China [36, 45]. Patients with diabetes develop retinopathy within the early stages of the disease, but this does not affect the sight unless the patient is treated, it progresses and eventually affects the sight [46].

Lifestyle intervention is remarkably effective in the primary prevention of diabetes and HTN. The initial approach to the management of both diabetes and HTN must emphasize weight control, regular physical activity, and dietary modification [47]. Further actions such as health education on a meal plan, treatment adherence, and blood glucose monitoring may improve blood glucose status.

The study has important limitations. Firstly, the study was conducted on both the type 1 and 2 diabetic populations. Secondly, the authors used specific factors to see their effect on diabetic retinopathy. Thirdly, the incidence of DR was not included in this study. Therefore, we recommended that further research need to be carried out on type 1 and 2 diabetic population separately, including other important factors. Moreover, the incidence of diabetic retinopathy need to be investigated based on the population characteristics.

## Implication of the study

The study's findings may result in a paradigm shift in the diabetic management process to reduce diabetic eye complications. As a result, health care providers, policymakers, and program planners could benefit from the study's findings. The study has an implication for clinicians' to provide important information about how effective a medical intervention to prevent the occurrence of diabetic retinopathy through early screening and treatment. The study also helps for decision and policy makers to plan and implement possible strategies to reduce the risks and progression of diabetic retinopathy.

## Conclusion

In this systematic and meta-analysis study, co-morbid HTN, poor glycemic control, and longer duration of diabetes illness were found to be the determinant factors of DR. Therefore, aggressive treatment of co-morbid HTN and blood glucose, and regular eye screening should be implemented to reduce the occurrence of DR among diabetic patients. In addition, healthcare workers should give due attention to those patients who have co-morbid HTN, poor glycemic control, and longer duration of diabetic illness.

## Supporting information

**S1 File. PRISMA checklist.**
(DOCX)

**S2 File. Data availability statement.**
(DOCX)

## Acknowledgments

We would like to acknowledge the team members for their invaluable contribution from the conception to the final approval for submission to publication.

## Author Contributions

**Conceptualization:** Abere Woretaw Azagew.

**Data curation:** Abere Woretaw Azagew, Zerko Wako Beko, Yohannes Mulu Ferede, Chilot Kassa Mekonnen.

**Formal analysis:** Abere Woretaw Azagew, Yeneabat Birhanu Yohanes, Zerko Wako Beko, Yohannes Mulu Ferede, Chilot Kassa Mekonnen.

**Investigation:** Abere Woretaw Azagew, Yeneabat Birhanu Yohanes, Zerko Wako Beko, Yohannes Mulu Ferede.

**Methodology:** Abere Woretaw Azagew, Yeneabat Birhanu Yohanes, Zerko Wako Beko, Yohannes Mulu Ferede, Chilot Kassa Mekonnen.

**Project administration:** Yeneabat Birhanu Yohanes.

**Software:** Abere Woretaw Azagew, Yeneabat Birhanu Yohanes, Zerko Wako Beko, Yohannes Mulu Ferede.

**Supervision:** Abere Woretaw Azagew, Zerko Wako Beko, Yohannes Mulu Ferede.

**Validation:** Abere Woretaw Azagew.

**Visualization:** Abere Woretaw Azagew, Yeneabat Birhanu Yohanes, Zerko Wako Beko, Yohannes Mulu Ferede.

**Writing – original draft:** Abere Woretaw Azagew.

**Writing – review & editing:** Abere Woretaw Azagew, Yeneabat Birhanu Yohanes, Zerko Wako Beko, Yohannes Mulu Ferede, Chilot Kassa Mekonnen.

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
