## [Decision Letter · Decision Letter 0]

22 Dec 2022

PONE-D-22-31170Determinants of Diabetic Retinopathy in Ethiopia; A systematic review and meta-analysisPLOS ONE

Dear Dr. Azagew,

Thank you for submitting your manuscript to PLOS ONE. After careful consideration, we feel that it has merit but does not fully meet PLOS ONE’s publication criteria as it currently stands. Therefore, we invite you to submit a revised version of the manuscript that addresses the points raised during the review process.

Please submit your revised manuscript by Feb 05 2023 11:59PM. If you will need more time than this to complete your revisions, please reply to this message or contact the journal office at plosone@plos.org. Please include the following items when submitting your revised manuscript:A rebuttal letter that responds to each point raised by the academic editor and reviewer(s). You should upload this letter as a separate file labeled 'Response to Reviewers'.A marked-up copy of your manuscript that highlights changes made to the original version. You should upload this as a separate file labeled 'Revised Manuscript with Track Changes'.An unmarked version of your revised paper without tracked changes. You should upload this as a separate file labeled 'Manuscript'.

We look forward to receiving your revised manuscript.

Kind regards,

Godwin Ovenseri-Ogbomo, OD, MPH, PhD

Academic Editor

PLOS ONE

Journal Requirements:

2. Please ensure you include the full search strategy such that it could be repeated. 

4. Please ensure that you refer to Figure 1 in your text as, if accepted, production will need this reference to link the reader to the figure.

5. Please upload a new copy of Figure 4 and 6 as the detail is not clear. Please follow the link for more information:

https://blogs.plos.org/plos/2019/06/looking-good-tips-for-creating-your-plos-figures-graphics/

https://blogs.plos.org/plos/2019/06/looking-good-tips-for-creating-your-plos-figures-graphics/

**Additional Editor Comments:**

Thank you for submitting your manuscript to PLOS ONE. I do concede the question raised need to be addressed. The manuscript needs to be reviewed and overhauled before it can be accepted for publication in the following areas:

1. The grammar needs to be reviewed. The generally acceptable tense for reporting research finding is the past tense. This was not the case in a substantial part of the manuscript.

2. I'm wondering why the author(s) only zero in on three determinants of DR in the manuscript. For a systematic review that seeks to provide evidence for evidence for practice, I think the net should have been cast more widely to consider variables such as age and sex of patient, HBA1C, family history of diabetes, weight/obesity, type of diabetes (T1 or T2), physical activity, occupation etc.

Reviewers' comments:

Reviewer's Responses to Questions

**Comments to the Author**

1. Is the manuscript technically sound, and do the data support the conclusions?

Reviewer #1: Yes

2. Has the statistical analysis been performed appropriately and rigorously? 

Reviewer #1: Yes

3. Have the authors made all data underlying the findings in their manuscript fully available?

Reviewer #1: Yes

4. Is the manuscript presented in an intelligible fashion and written in standard English?

Reviewer #1: No

5. Review Comments to the Author

Reviewer #1: Manuscript Number: PONE-D-22-31170

Manuscript Title: Determinants of Diabetic Retinopathy in Ethiopia; A systematic review and meta-analysis

Congratulations dear authors on your scholarly work; you have brought an important study problem with good findings that have public health importance in the area of practice. However, there are methodological and other concerns that I want you to address before considering the manuscript for publication.

General comment

There are multiple typological and grammar usage errors that need extensive proof reading for revisions.

Specific comments

Abstract

1. “Background” should state burden of diabetic retinopathy in Ethiopia. i.e about one fifth of the diabetic patients, 19.48%.

2. Statistics is ok and well described.

3. Conclusions are supported by the findings.

Introduction:

1. There are redundant ideas and hence the introduction needs further synthesis.

2. What were the unanswered questions, and what will you add to your research?

3. Finally, please include your rationale for doing this research paper.

Methods

1. Is the protocol of this review registered? If so, include its registration number. If not, state its current status.

2. I suggest the authors consider using a PICO (possibly with the T or S) format to make the review question explicit and assist with clear specification of the inclusion and exclusion criteria.

3. Provide the primary search string, including the truncation and synonyms as a supplementary file for at least one database.

4. Specify the type of searching strategy (e.g. was it line by line, by combined concepts, did the search include title (TI), abstract (Ab) or full text or all these categories) was used, specify if databases were searched independently and if any modifications were made to the search strategy (e.g. limiters) for different databases. The description of the method should include enough details for reproducibility. Explain the steps in screening (e.g. title, abstract, full text).

5. I suggest reporting the statistics for measurement of the level of agreement for the independent reviews (e.g. quality scores) of each article. Moreover, kindly append a table showing methodological quality of the appraised articles with the last column being ‘overall quality score’.

6. How did you pool the individual knowledge, attitude and practice scores because nearly all the primary studies used to consider their own operational definitions? Please be critical in this issue and support your measurement of these pooled outcomes with sound references.

7. Explain the data extracted from the studies, reliability and validity of data extraction tool. Explain how the variables extracted were determined and criteria for data to be suitable for extraction. Explain if data transformation was required or undertaken when data were reported differently.

Results

Please include two columns in Table 1: data collection technique (interview, observation, self administered questionnaire, etc) and funding source for each study (you can say not funded, not reported or name of funder if funded).

Discussion

1. The discussion needs to be re-thought as it provides certain explanations that may be inaccurate.

2. There has to be a separate subsection named as “implication of the study” showing beneficence of the findings to the clinicians, decision makers and policy makers.

3. Please include a detailed analysis of the limitations that users should be mindful of interpreting the findings.

6. PLOS authors have the option to publish the peer review history of their article (what does this mean?). If published, this will include your full peer review and any attached files.

Reviewer #1: 

---

## [Author Response · Author response to Decision Letter 0]

23 Jan 2023

The response to reviewers' are attached at "attach files" section.

---

## [Editor Report · Decision Letter 1]

10 Apr 2023

PONE-D-22-31170R1Determinants of Diabetic Retinopathy in Ethiopia: A systematic review and meta-analysisPLOS ONE

Dear Dr. Azagew,

Thank you for submitting your manuscript to PLOS ONE. After careful consideration, we feel that it has merit but does not fully meet PLOS ONE’s publication criteria as it currently stands. Therefore, we invite you to submit a revised version of the manuscript that addresses the points raised during the review process.

Having gone through your reviewed manuscript vis-à-vis the reviewer's comments, I'm afraid there are a few issues that needs your attention.

1. State the research question using PICO format.

2. Address the issue whether a protocol for the systematic review was registered and if so report the registration details in the manuscript.

3. In line 246 of the revised manuscript, please replace "don't" with "do not".

4. Rework the implication of the study to indicate clearly what the implications of the results are for clinicians, patients and decision makers.

We look forward to receiving your revised manuscript.

Kind regards,

Godwin Ovenseri-Ogbomo, OD, MPH, PhD

Academic Editor

PLOS ONE

Journal Requirements:

Additional Editor Comments:

Thank you for sending in your revised manuscript.

Having gone through your reviewed manuscript vis-à-vis the reviewer's comments, I'm afraid there are a few issues that needs your attention.

1. State the research question using PICO format.

2. Address the issue whether a protocol for the systematic review was registered and if so report the registration details in the manuscript.

3. In line 246 of the revised manuscript, please replace "don't" with "do not".

4. Rework the implication of the study to indicate clearly what the implications of the results are for clinicians, patients and decision makers.
---

## [Author Response · Author response to Decision Letter 1]

12 May 2023

The response to reviewers are attached at file attachment section.

---

## [Editor Report · Decision Letter 2]

22 May 2023

Determinants of diabetic retinopathy in Ethiopia: A systematic review and meta-analysis

PONE-D-22-31170R2

Dear Dr. Azagew,

We’re pleased to inform you that your manuscript has been judged scientifically suitable for publication and will be formally accepted for publication once it meets all outstanding technical requirements.

Kind regards,

Godwin Ovenseri-Ogbomo, OD, MPH, PhD

Academic Editor

PLOS ONE

Additional Editor Comments (optional):

Thank you for submitting the revised manuscript and for taking on board the reviewers' comments and suggestions.
---

## [Editor Report · Acceptance letter]

31 May 2023

PONE-D-22-31170R2 

Determinants of diabetic retinopathy in Ethiopia: A systematic review and meta-analysis 

Dear Dr. Azagew:

I'm pleased to inform you that your manuscript has been deemed suitable for publication in PLOS ONE. Congratulations! Your manuscript is now with our production department. 

Kind regards, 

on behalf of

Dr. Godwin Ovenseri-Ogbomo 

Academic Editor

PLOS ONE